# Comparative Prospective and Longitudinal Analysis on the Platelet-to-Lymphocyte, Neutrophil-to-Lymphocyte, and Albumin-to-Globulin Ratio in Patients with Non-Metastatic and Metastatic Prostate Cancer

Stefano Salciccia [1], Marco Frisenda [1], Giulio Bevilacqua [1], Pietro Viscuso [1], Paolo Casale [2], Ettore De Berardinis [1], Giovanni Battista Di Pierro [1], Susanna Cattarino [1], Gloria Giorgino [1], Davide Rosati [1], Francesco Del Giudice [1], Antonio Carbone [3], Antonio Pastore [3], Benjamin I. Chung [4], Michael L. Eisenberg [4], Riccardo Autorino [5], Simone Crivellaro [6], Flavio Forte [7], Alessandro Sciarra [1,*], Gianna Mariotti [1] and Alessandro Gentilucci [1]

1   Department of "Materno Infantile e Scienze Urologiche, 'Sapienza' University of Rome, Policlinico Umberto I Hospital, 00161 Rome, Italy
2   Department of Urology, Humanitas, 20089 Milan, Italy
3   Unit of Urology, ICOT Hospital Latina, Sapienza University, 04100 Latina, Italy
4   Department of Urology, Stanford University School of Medicine, Stanford, CA 94305, USA
5   Department of Urology, Rush University Medical Center, Chicago, IL 60612, USA
6   Department of Urology, University of Illinois Hospital e Camp, Health Sciences System, Chicago, IL 60612, USA
7   Urology Department, M.G. Vannini Hospital, 00177 Rome, Italy
*   Correspondence: alessandro.sciarra@uniroma1.it

**Abstract:** Purpose: To prospectively evaluate the albumin/globulin ratio (AGR), neutrophil/lymphocyte ratio (NLR), and platelet/lymphocyte ratio (PLR) diagnostic and prognostic predictive value in a stratified population of prostate cancer (PC) cases. Methods: Population was divided based on the clinical and histologic diagnosis in: Group A: benign prostatic hyperplasia (BPH) cases (494 cases); Group B: all PC cases (525 cases); Group B1: clinically significant PC (426 cases); Group B2: non-metastatic PC (416 cases); Group B3: metastatic PC (109 cases). NLR, PLR, and AGR were obtained at the time of the diagnosis, and only in cases with PC considered for radical prostatectomy, determinations were also repeated 90 days after surgery. For each ratio, cut-off values were determined by receiver operating characteristics curve (ROC) analysis and fixed at 2.5, 120.0, and 1.4, respectively, for NLR, PLR, and AGR. Results: Accuracy in predictive value for an initial diagnosis of clinically significant PC (csPC) was higher using PLR (0.718) when compared to NLR (0.220) and AGR (0.247), but, despite high sensitivity (0.849), very low specificity (0.256) was present. The risk of csPC significantly increased only according to PLR with an OR = 1.646. The percentage of cases with metastatic PC significantly increased according to high NLR and high PLR. Accuracy was 0.916 and 0.813, respectively, for NLR and PLR cut-off, with higher specificity than sensitivity. The risk of a metastatic disease increased 3.2 times for an NLR > 2.5 and 5.2 times for a PLR > 120 and at the multivariate analysis. Conclusion: PLR and NLR have a significant predictive value towards the development of metastatic disease but not in relation to variations in aggressiveness or T staging inside the non-metastatic PC. Our results suggest an unlikely introduction of these analyses into clinical practice in support of validated PC risk predictors.

**Keywords:** albumin-to-globulin ratio; neutrophil-to-lymphocyte ratio; metastatic; platelet-to-lymphocyte ratio; prostatic neoplasm; radical prostatectomy

## 1. Introduction

Prostate cancer (PC) is an extremely heterogeneous tumor and clinical decisions continue to depend upon serum prostate-specific antigen (PSA) levels, tumor stage, risk

classes, and Gleason score [1,2]. Predictive nomograms mainly including these clinical parameters are also used to evaluate the risk of advanced stage, undifferentiated tumors, and progression after treatments [3,4].

Different research sustains the hypothesis that chronic inflammation and the immune environment can condition carcinogenesis and tumor progression. The neutrophil-to-lymphocyte ratio (NLR) and platelet-to-lymphocyte ratio (PLR) can be easily obtained from routine blood counts and they have been proposed as markers of the relationship between inflammation or immune responses and tumor growth or progression [5,6]. Low lymphocyte counts and increased platelet counts have been associated with adverse prognostic features for different diseases, including PC [7]. Additionally, hypoalbuminemia can be associated with systemic inflammation in patients with cancer [8]. Inflammatory reaction and immunity are influenced by serum albumin and globulin; hypoalbuminemia and hyperglobulinemia are considered indicators of chronic inflammation in oncologic patients [8,9]. Albumin can reflect the body's nutritional status, globulin the immunological and inflammatory status, and their ratio can be evaluated as albumin divided by total protein minus albumin value in serum [10].

A prognostic role for NLR and PLR has been underlined for several solid tumors [8]. In PC the significance of NLR or PLR has been investigated in different settings, more frequently in advanced metastatic PC submitted to systemic therapies. Most clinical trials on NLR and PLR in PC are retrospective and different meta-analysis [9–11] showed a high level of heterogeneity of results among populations and studies, suggesting that either a high PLR or a high NLR are correlated with poor prognosis in PC [9,10]. Similarly, most clinical trials on AGR in PC are retrospective and different meta-analyses [8,10] showed contrasting results regarding a predictive value for low preoperative AGR in terms of poor prognosis in PC.

Now there has been relevant interest in NLR, PLR, and AGR in PC but data still remain controversial, with mainly retrospective analysis on advanced disease and heterogeneous non-stratified populations.

*Aim and Objectives*

The aim of the present analysis is to prospectively evaluate and compare the AGR, NLR, and PLR diagnostic and prognostic predictive value in a population of PC cases in comparison with benign prostatic hyperplasia (BPH) patients. In particular, we compared the predictive value of the three ratios either in terms of initial diagnosis of PC or in terms of advanced local staging, systemic metastases, or undifferentiated ISUP grading. Moreover, in a subpopulation of non-metastatic PC cases considered for radical prostatectomy (RP), we longitudinally analyzed AGR, PLR, and NLR variations after surgery and in relation to PSA progression.

## 2. Materials and Methods

### 2.1. Study Design

This is a prospective, longitudinal, and mono-center study. From January 2021 to August 2022, patients were consecutively enrolled as outpatients referred to our clinic for the management of prostatic diseases. A real-life situation is analyzed, and all diagnostic and therapeutic procedures reflected our routine clinical practice in a department at high volume for the management of PC disease following recommendations of the European Association of Urology (EAU) guidelines. The protocol was approved by our internal ethical committee and all patients gave their informed consensus for each analysis. In all cases, AGR, NLR, and PLR determination were obtained at baseline when the diagnosis was defined. In non-metastatic cases considered for radical prostatectomy (RP), after discussion of treatment options and presentation to the patient, the ratios were obtained either at baseline or after RP.

*2.2. Population*

The population was divided based on the clinical and histologic diagnosis in: Group A: BPH cases; Group B: PC cases; Group B1: clinically significant PC; Group B2: non-metastatic PC; Group B3: metastatic PC.

Inclusion criteria were: Group A: new histologic diagnosis of BPH and/or clinical diagnosis of BPH without evidence or suspicious for PC; prostate volume > 30 cc, IPSS > 7, PSA level $\leq$ 2.5 ng/mL or if >2.5 ng/mL not suspicious (PIRADS 1–2) for PC at multiparametric magnetic resonance (mMR) and no evidence for PC at biopsy; Group B: new histologic diagnosis of prostatic adenocarcinoma at biopsy; Group B1: new histologic diagnosis of prostatic adenocarcinoma at biopsy, clinically significant as defined by ISUP grading > 1; Group B2: new histologic diagnosis of prostatic adenocarcinoma at biopsy, no evidence of distant metastasis at systemic imaging; Group B3: new histologic diagnosis of prostatic adenocarcinoma at biopsy, at least one distant metastasis at systemic imaging, stratified in oligometastatic (less than four distant metastasis) and poli-metastatic (four or more distant metastasis).

Exclusion criteria were previous or actual androgen deprivation therapies, chemotherapies, immunotherapies, pelvic radiation therapies, treatments with other agents that could influence prostate growth and immune system, and actual diagnosis of infections, and inflammation or immunity disorders.

*2.3. Methods*

All cases were submitted to diagnostic and therapeutic practices reflecting our routine clinical activity and following EAU guidelines for the initial diagnosis and management of BPH and PC cases. In particular, either in BPH or in PC cases, prostate volume was assessed using the ellipsoid evaluation at ultrasonography. In cases with suspicion of PC, mMR was performed, a PIRADS v2 score was defined and in cases with PIRADS score 3–5, a standard 12-core random biopsy was associated with targeted samples on the sites indicated by mMR. In cases with a histologic diagnosis of PC at biopsy, clinical staging and risk category (D'Amico and EAU classification) assessment was homogeneously performed following EAU guidelines. In particular, local staging was obtained at mMR and systemic staging using bone scan and CT scan or PET-CT scan.

2.3.1. Treatment Choice in Prostate Cancer Cases

In all cases with a new PC diagnosis, treatment decision was considered on the basis of risk classes determination and staging according to EAU guidelines after discussion of the different options with the patient. In particular, in patients considered for RP, every procedure was performed using a standard robotic-assisted (RARP) or laparoscopic (LRP) intraperitoneal approach consistent with best practice. Extended lymph node dissection (eLND) was performed in all cases with high risk or intermediate risk and more than 5% expected risk for positive lymph nodes at Briganti nomogram, including bilateral removal of the nodes overlying the external iliac artery and vein, the nodes within obturator fossa, and the nodes medial and lateral to the internal iliac artery.

2.3.2. Pathologic Evaluation

All histologic specimens from prostatic biopsy and RP were analyzed by a uropathologist with a long experience in the PC field. Prostatic adenocarcinoma diagnosis was associated with the determination of ISUP grading, percentage of positive samples for PC, and maximal percentage of PC tissue per core at biopsy and prostate tumor volume, pathologic T and N staging, surgical margin status, presence of perineural invasion (PNI), cribriform, and intraductal (IDC) differentiation at RP.

2.3.3. Neutrophil-to-Lymphocyte, Platelet-to-Lymphocyte, and Albumin-to-Globulin Ratio Determination

In all cases for each Group, NLR, PLR, and AGR were obtained at baseline at the time of the diagnosis of BPH or PC using results from routine blood count and proteinogram. Only in cases with PC considered for RP, NLR, PLR, and AGR determinations were repeated 90 days after surgery, using a post-operative laboratory routine control.

For each ratio, cut-off values were determined by receiver operating characteristics curve (ROC) analysis using Youden's index [12], and the optimal cut-off in our population was 2.5, 120.0, and 1.4, respectively, for NLR, PLR, and AGR, so as to distinguish low and high rate cases in each Group.

*2.4. Statistical Analysis*

Calculations were accomplished using Stata version 1.7 (Stata Corporation, College Station, TX, USA) with all tests being two-sided, and statistical significance set at <0.05.

For the comparison of quantitative data and pairwise intergroup comparisons of variables, a Mann–Whitney test was performed. For the comparison of qualitative data, Fisher's Exact test and chi-square test were used. Pearson correlation analysis was also performed. Univariate and multivariate Cox proportional analyses considering clinical and pathological parameters were used. We tested and compared the accuracy of the AGR, NLR, and PLR for predicting either the initial diagnosis of PC or its staging and aggressiveness. Regression coefficients were used to calculate the risk according to each model and the discrimination accuracy of these models was quantified using the area under the receiver operating characteristic (ROC) curve (AUC). Sensitivity, specificity, positive predictive value (PPV), and negative predictive value (NPV) of the different ratios in predicting PC diagnosis and adverse staging or grading were evaluated. HRs and corresponding 95% CI at univariate and multivariate analysis were considered to evaluate the importance and independency of the prognostic value for the different ratios.

**3. Results**

A total of 1019 consecutive cases responded to our inclusion and exclusion criteria and were enrolled in the analysis. Table 1 shows the clinical and pathological characteristics of our population stratified into different groups. In particular, 494 cases were enrolled in Group A as BPH cases and 525 cases in Group B as PC cases. PC cases were further stratified into clinically significant (426 cases in Group B1), non-metastatic (416 cases in Group B2), and metastatic (109 cases in Group B3).

**Table 1.** Characteristics of the whole population included in the study. Group A = patients with diagnosis of benign prostatic hyperplasia (BPH). Group B = patients with diagnosis of prostate cancer (PC), clinically significant in Group B1, non-metastatic (nmPC) in Group B2, and metastatic (mPC) in Group B3. Mean $\pm$ SD, median, (range). Number of cases (%).

| Parameter | Group A (BPH) | Group B (all PC) | Group B1 (Clinically Significant PC) | Group B2 (nmPC) | Group B3 (mPC) | *p* Value 1 = A vs. B 2 = A vs. B1 3 = A vs. B2 4 = A vs. B3 5 = B2 vs. B3 |
|---|---|---|---|---|---|---|
| Number of cases | 494 | 525 | 426 | 416 | 109 | \ |
| Age (years) | 66.3 $\pm$ 9.2; 68.0: (23–87) | 66.6 $\pm$ 8.5; 67.0: (40–89) | 65.1 $\pm$ 8.5; 67.0 (44–84) | 65.3 $\pm$ 8.4; 67.0 (40–87) | 71.2 $\pm$ 8.4; 67.0 (52–89) | 1. 0.591 2. 0.044 3. 0.086 4. <0.0001 5. <0.0001 |
| BMI | 25.2 $\pm$ 8.7; 25.0: (16.9–45.6) | 26.0 $\pm$ 3.3; 25.3 (16.9–39.4) | 27.1 $\pm$ 7.6; 25.4 (20.4–39.4) | 26.1 $\pm$ 3.3; 25.4 (16.9–39.4) | 25.4 $\pm$ 3.3; 25.3 (22.7–29.7) | 1. 0.0504 2. 0.0007 3. 0.0367 4. 0.8301 5. 0.0259 |
| Metabolic syndrome % 0 (absent) 1 (mild) 2 (complete) | 83.6% 8.7% 7.7% | 64.8% 17.3% 17.9% | 66% 16.7% 17.3% | 63.9% 19.6% 16.5% | 65.1% 10.7% 24.2% | 1. <0.0001 2. <0.0001 3. <0.0001 4. <0.0001 5. <0.0001 |
| Baseline NLR  Low (2.5) High (2.5) | 2.2 $\pm$ 1.2; 2.1: (0.5–17.8) (data on 459 cases) 327 (71.2%) 132 (28.8%) | 2.5 $\pm$ 1.1; 2.2: (0.1–9.6) (data on 455 cases) 259 (56.9%) 196 (43.1%) | 2.4 $\pm$ 1.1; 2.2: (0.1–8) (data on 371 cases) 205 (55.2%) 166 (44.8%) | 2.5 $\pm$ 1.1; 2.2: (0.1–9.6) (data on 355 cases) 220 (61.6%) 135 (38.4%) | 2.6 $\pm$ 1.5; 2.2 (0.6–4.6) (data on 100 cases) 39 (40.7%) 61 (59.3%) | 1. 0.0005 2. 0.0053 3. 0.0038 4. 0.0019 5. 0.1528 |
| Baseline PLR  Low (<120.0) High (120.0) | 117.3 $\pm$ 51.3; 113.0: (28.3–369.5) (data on 456 cases) 289 (63.3%) 167 (36.7%) | 132.8 $\pm$ 49.3; 116.0: (1.92–439.0) (data on 455 cases) 193 (42.4%) 262 (57.6%) | 129.2 $\pm$ 49.3; 116.3: (1.92–439.0) (data on 297 cases) 149 (50.1%) 148 (49.9%) | 127.8 $\pm$ 49.3; 116.3: (1.9–439.0) (data on 355 cases) 173 (49.1%) 182 (50.9%) | 149.6 $\pm$ 49.5; 116.3 (58.8–255.4) (data on 100 cases) 20 (19.4%) 80 (80.6%) | 1. <0.0001 2. <0.0001 3. <0.0001 4. <0.0001 5. <0.0001 |

**Table 1.** *Cont.*

| Parameter | Group A (BPH) | Group B (all PC) | Group B1 (Clinically Significant PC) | Group B2 (nmPC) | Group B3 (mPC) | *p* Value<br>1 = A vs. B<br>2 = A vs. B1<br>3 = A vs. B2<br>4 = A vs. B3<br>5 = B2 vs. B3 |
|---|---|---|---|---|---|---|
| Baseline AGR | 1.5 ± 0.3; 1.5: (0.7–2.1)<br>(data on 239 cases) | 1.5 ± 0.3; 1.5: (0.8–5.6)<br>(data on 420 cases) | 1.5 ± 0.3; 1.5: (0.9–5.6)<br>(data on 344 cases) | 1.5 ± 0.3; 1.5: (0.9–5.6)<br>(data on 327 cases) | 1.5 ± 0.3; 1.5 (0.8–1.9)<br>(data on 93 cases) | 1. 0.1923<br>2. 0.4036<br>3. 0.2164<br>4. 0.2499<br>5. 0.7587 |
| Low (≤1.4) | 52 (21.7%) | 149 (35.4%) | 127 (36.9%) | 129 (39.8%) | 20 (20.8%) | |
| High (>1.4) | 187 (78.3%) | 271 (64.6%) | 217 (63.1%) | 198 (60.2%) | 73 (79.2%) | |
| Prostate volume (cc) | 51.5 ± 18.9; 46.0: (25–200) | 46.0 ± 18.7; 45.0: (14–104) | 42.8 ± 20.3; 45.0 (14–86) | 45.1 ± 20.2; 45.0 (14–104) | 50.0 ± 20.3; 45.0 (32–90) | 1. <0.0001<br>2. <0.0001<br>3. <0.0001<br>4. 0.4669<br>5. 0.0217 |
| total PSA (ng/mL) | 3.2 ± 12.5; 3.2: (0.2–7.3) | 13.3 ± 11.7; 4.6: (1.7–106.0) | 14.7 ± 11.8; 4.7 (1.7–86.0) | 9.3 ± 11.8; 4.7 (1.7–86.0) | 26.5 ± 11.9; 4.5 (0.2–106.0) | 1. <0.0001<br>2. <0.0001<br>3. <0.0001<br>4. <0.0001<br>5. <0.0001 |
| PSAD | 0.05 ± 0.03; 0.04: (0.004–0.09) | 0.36 ± 0.24; 0.08: (0.0025–2.2) | 0.34 ± 0.24; 0.08 (0.01–1.56) | 0.22 ± 0.24; 0.08 (0.01–1.56) | 0.54 ± 0.24; 0.08 (0.01–2.20) | 1. <0.0001<br>2. <0.0001<br>3. <0.0001<br>4. <0.0001<br>5. <0.0001 |
| mMR PIRADS score | \ | (data on 257) | (data on 194) | (data on 252) | | 1. \ <br>2. \ <br>3. \ <br>4. \ <br>5. \ |
| PIRADS 2 | | 12 (4.5%) | 9 (4.7%) | 12 (4.8%) | - | |
| PIRADS 3 | | 36 (14.3%) | 24 (12.3%) | 35 (13.9%) | - | |
| PIRADS 4 | | 153 (59.0%) | 115 (59.3%) | 151 (59.9%) | - | |
| PIRADS 5 | | 56 (22.2%) | 46 (23.7%) | 54 (21.4%) | - | |

**Table 1.** *Cont.*

| Parameter | Group A (BPH) | Group B (all PC) | Group B1 (Clinically Significant PC) | Group B2 (nmPC) | Group B3 (mPC) | p Value 1 = A vs. B 2 = A vs. B1 3 = A vs. B2 4 = A vs. B3 5 = B2 vs. B3 |
|---|---|---|---|---|---|---|
| Prostate tumor size (mm) at MR | \ | 12.3 ± 4.8; 12.0: (4–35) | 12.7 ± 4.9; 12.0 (4–52) | 12.2 ± 4.9; 12.0 (4–35) | 15.8 ± 5.0; 12.0 (18–25) | 1. \ 2. \ 3. \ 4. \ 5. <0.0001 |
| Clinical T staging T2 T3a T3b | \ | 454 (86.5%) 62 (11.8%) 9 (12.7%) | 356 (83.6%) 62 (14.6%) 8 (1.8%) | 365 (88.1%) 42 (9.9%) 9 (2%) | 89 (80%) 20 (20%) 0 (0%) | 1. \ 2. \ 3. \ 4. \ 5. <0.0001 |
| Clinical N staging N0 N1 | \ | 447 (85.1%) 78 (14.9%) | 350 (82.1%) 76 (17.9%) | 390 (94.2%) 26 (5.8%) | 57 (51.7%) 52 (48.3%) | 1. \ 2. \ 3. \ 4. \ 5. <0.0001 |
| M staging M0 M1 oligometastatic (<4) M1 polimetastatic (≥4) | \ | 413 (78.6%) 106 (20.2%) 6 (1.2%) | 314 (73.7%) 106 (24.8%) 6 (1.5%) | 416 (100%) 0 (0%) 0 (0%) | 0 (0%) 103 (94.6%) 6 (5.4%) | 2. \ 3. \ 4. \ 5. <0.0001 |
| Biopsy outcomes % positive samples PC Max% PC tissue per core | \ | 35.4 ± 26.2; 28.0: (4.0–100.0) 40.2 ± 25.8; 35.0: (2.0–94.0) | 40.2 ± 26.3; 28.0: (4.0–100.0) 43.9 ± 25.9; 35.0: (4.0–94.0) | 34.4 ± 26.3; 28.0 (4.0–100.0) 35.0 ± 25.9; 35.0: (2.0–94.0) | 41.7 ± 26.3; 28.0 (50.0–100.0) 63.2 ± 26.1; 35.0 (32.0–94.0) | 1. \ 2. \ 3. \ 4. \ 5. 0.0097 |

**Table 1.** *Cont.*

| Parameter | Group A (BPH) | Group B (all PC) | Group B1 (Clinically Significant PC) | Group B2 (nmPC) | Group B3 (mPC) | *p* Value 1 = A vs. B 2 = A vs. B1 3 = A vs. B2 4 = A vs. B3 5 = B2 vs. B3 |
|---|---|---|---|---|---|---|
| ISUP grading at biopsy | | | | | | |
| 1 | | 98 (18.6%) | 0 (0%) | 98 (23.7%) | 0 (0%) | 1. \ |
| 2 | | 171 (32.6%) | 170 (39.9%) | 170 (40.9%) | 1 (1.8%) | 2. \ |
| 3 | \ | 107 (20.4%) | 107 (25.1%) | 77 (18.7%) | 30 (26.8%) | 3. \ |
| 4 | | 113 (21.5%) | 113 (26.6%) | 46 (10.7%) | 67 (61.6%) | 4. \ |
| 5 | | 36 (6.9%) | 36 (8.4%) | 25 (6%) | 11 (9.8%) | 5. 0.08 |
| Risk Class (D'Amico) | | | | | | |
| Low risk | | 104 (19.8%) | 17 (3.9%) | 104 (25.2%) | - | |
| Intermediate risk | \ | 219 (41.8%) | 210 (49.3%) | 219 (52.8%) | - | \ |
| High risk | | 202 (38.4%) | 199 (46.8%) | 93 (22%) | - | |
| Radical prostatectomy | | (Data on 371) | (Data on 281) | (Data on 371) | | |
| Laparoscopic | \ | 223 (60.1%) | 166 (59%) | 223 (60.1%) | \ | \ |
| Robotic-assisted | | 148 (39.9%) | 115 (41%) | 148 (39.9%) | | |
| Pathological stage (T) | | (Data on 371) | (Data on 281) | (Data on 371) | | |
| pT2 | | 190 (51.2%) | 112 (39.9%) | 190 (51.2%) | | |
| pT3a | \ | 141 (38%) | 130 (46.3%) | 141 (38%) | \ | \ |
| pT3b | | 40 (10.8%) | 39 (13.8%) | 40 (10.8%) | | |
| pT4 | | 0 (0%) | 0 (0%) | 0 (0%) | | |
| Pathological stage (N) | | | | | | |
| N0 | \ | 284 (94%) | 213 (92.2%) | 284 (94%) | \ | \ |
| N+ | | 18 (6%) | 18 (1.8%) | 18 (6%) | | |

**Table 1.** *Cont.*

| Parameter | Group A (BPH) | Group B (all PC) | Group B1 (Clinically Significant PC) | Group B2 (nmPC) | Group B3 (mPC) | *p* Value 1 = A vs. B 2 = A vs. B1 3 = A vs. B2 4 = A vs. B3 5 = B2 vs. B3 |
|---|---|---|---|---|---|---|
| ISUP grading at surgery | | (data on 371) | (data on 281) | (data on 371) | | |
| 1 | | 64 (17.4%) | 4 (1.4%) | 64 (17.3%) | | |
| 2 | | 163 (43.9%) | 140 (49.8%) | 163 (43.9%) | | |
| 3 | \ | 85 (23%) | 79 (28.2%) | 85 (23%) | \ | \ |
| 4 | | 23 (6.2%) | 23 (8.1%) | 23 (6.3%) | | |
| 5 | | 36 (9.5%) | 35 (12.4%) | 36 (9.5%). | | |
| Surgical margin (R) | | | | | | |
| Negative | \ | 280 (78.2%) | 209 (74.3%) | 280 (78.2%) | \ | \ |
| Positive | | 91 (21.8%) | 72 (25.7%) | 91 (21.8%) | | |
| PNI at surgery | | | | | | |
| Positive | \ | 258 (62.2%) | 213 (75.8%) | 258 (62.2%) | \ | \ |
| Negative | | 113 (37.8%) | 68 (24.2%) | 113 (37.8%) | | |
| Cribriform/IDC at surgery | | | | | | |
| Positive | \ | 68 (16.4%) | 57 (20.2%) | 68 (16.5%) | \ | \ |
| Negative | | 303 (83.6%) | 224 (79.8%) | 303 (83.5%) | | |
| Postoperative total PSA (ng/mL) | \ | 0.2 ± 0.9; 0.02: (0.01–10) | 0.2 ± 0.9; 0.02: (0.01–10) | 0.2 ± 0.9; 0.02: (0.01–10) | \ | \ |
| Biochemical progression (number of cases and %) | \ | 46 (12.3%) | 41 (14.5%) | 46 (12.3%) | \ | \ |

### 3.1. Comparative Analysis among the Different Groups

Table 1 shows the distribution of the different variables according to the different groups examined and Figure 1 shows the distribution of low and high ratios according to the diagnosis in the different groups.

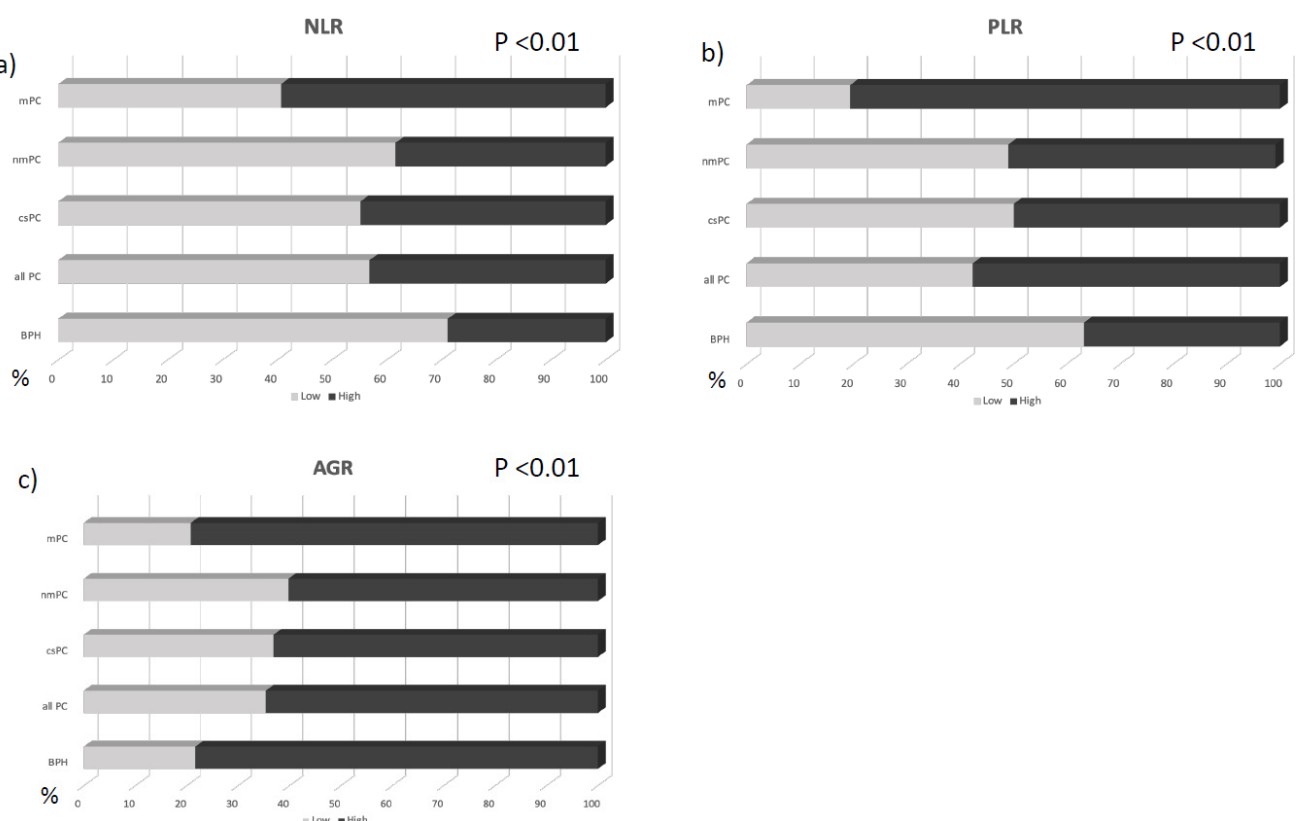

**Figure 1.** Percentage of cases with low or high NLR (**a**), PLR (**b**), and AGR (**c**) according to the diagnosis in the different groups.

#### 3.1.1. Benign Prostatic Hyperplasia versus Prostate Cancer Cases

No significant differences ($p = 0.591$) were found between Group A (BPH cases) and Group B (PC cases) in terms of age, whereas total PSA was significantly higher ($p < 0.001$) in Group B than in Group A. Mean values of NLR and PLR were significantly ($p < 0.001$) lower in Group A ($2.23 \pm 1.22$ and $117.30 \pm 51.3$, respectively) than in Group B ($2.49 \pm 1.14$ and $132.8 \pm 49.3$, respectively) and the percentage of cases with high NLR and PLR according to the cut-off, was higher in Group B (43.1% and 57.6%, respectively) than in Group A (28.8% and 36.7%, respectively). On the contrary, no significant differences ($p = 0.1923$) in mean values of AGR were found between Group A ($1.52 \pm 0.29$) and Group B ($1.49 \pm 0.28$) but the percentage of cases with low AGR was higher in Group B (35.4%) than in Group A (21.7%) (Table 1).

Considering Group B1 (clinically significant PC), mean values of NLR and PLR ($2.45 \pm 1.15$ and $129.19 \pm 49.32$, respectively) were significantly ($p = 0.0053$) higher when compared with Group A and the percentage of cases with high NLR and PLR was higher in Group B1 (44.8% and 49.9%, respectively) than in Group A (Table 1). On the contrary, no significant differences ($p = 0.4036$) in mean values of AGR were found between Group B1 ($1.50 \pm 0.28$) and Group A but the percentage of cases with low AGR was higher in Group B1 (36.9%) than in Group A (Table 1).

### 3.1.2. Non-Metastatic versus Metastatic Prostate Cancer Cases

In Group B3, 94.6% of cases were oligometastatic and only 5.4% were poli-metastatic. Mean values of PLR were significantly ($p < 0.001$) higher in metastatic (Group B3: $149.63 \pm 49.46$) than in non-metastatic (Group B2: $127.79 \pm 49.32$) cases. On the contrary, no significant differences ($p > 0.05$) between Group B2 and Group B3 were found in terms of mean values of PLR and AGR (Table 1). The percentage of cases with high NLR and PLR according to the cut-off was higher in Group B3 (59.3% and 80.6%, respectively) than in Group B2 (38.4% and 50.9%, respectively) whereas no significant variations between Group B2 and Group B3 were found in terms of low AGR distribution (Table 1).

### 3.2. Results on the Basis of Neutrophil-to-Lymphocyte, Platelet-to-Lymphocyte, and Albumin-to-Globulin Ratio Stratification (Low versus High)

Tables 2 and 3 show the distribution of the different clinical and pathological variables according to the stratification of cases on the basis of NLR, PLR, and AGR cut-offs.

**Table 2.** Characteristics of the population stratified on the basis of NLR, PLR, and AGR. Mean $\pm$ SD, median, (range). Number of cases (%).

| Parameter | Low NLR (<2.5) | High NLR ($\geq$2.5) | Low PLR (<120.0) | High PLR ($\geq$120.0) | Low AGR (<1.4) | High AGR ($\geq$1.4) | *p* Value 1. NLR 2. PLR 3. AGR |
|---|---|---|---|---|---|---|---|
| Number of cases | 586 | 328 | 482 | 429 | 201 | 458 | / |
| Age (years) | 66.1 ± 8.7 66.0 (40–89) | 66.8 ± 7.9 68.0 (44–85) | 66.2 ± 8.7 67.0 (40–89) | 66.4 ± 8.3 67.5 (23–85) | 67.9 ± 6.9 68.4 (48.84) | 65.7 ± 8.4 67.3 (23–87) | 1. 0.2347 2. 0.7246 3. 0.0856 |
| BMI | 25.7 ± 3.4 26.4 (16.9–45.6) | 25.5 ± 2.9 25.4 (16.9–41.5) | 25.9 ± 3.6 25.5 (17.0–45.6) | 25.3 ± 2.8 25.2 (16.9–41.5) | 26.2 ± 3.7 25.6 (18.5–41.5) | 25.8 ± 3.0 25.5 (16.9–36.3) | 1. 0.3099 2. 0.0044 3. 0.1263 |
| Metabolic syndrome | | | | | | | 1. 0.522 2. 0.890 3. 0.350 |
| 0 (absent) | 432 (73.72%) | 237 (72.26%) | 346 (71.78%) | 320 (74.59%) | 127 (63.18%) | 326 (71.18%) | |
| 1 (mild) | 77 (13.14%) | 44 (13.41%) | 64 (13.28%) | 57 (13.29%) | 37 (18.41%) | 60 (13.10%) | |
| 2 (complete) | 77 (13.14%) | 47 (14.33%) | 72 (14.94%) | 52 (12.12%) | 37 (18.41%) | 72 (15.72%) | |
| Prostate volume (cc) | 50.0 ± 22.0 46 (14–274) | 47.9 ± 17.5 45 (14–165) | 50.5 ± 23.3 45 (14–274) | 47.8 ± 16.8 45 (14–165) | 49.9 ± 21.2 45 (15–165) | 50.0 ± 22.3 47 (14–274) | 1. 0.1304 2. 0.0515 3. 0.9228 |
| Total PSA (ng/mL) | 7.6 ± 11.3 4.2 (0.05–106.0) | 10.4 ± 12.8 5.8 (0.06–97.0) | 6.9 ± 10.7 3.9 (0.05–105.0) | 10.4 ± 12.9 6.0 (0.06–106.0) | 10.3 ± 10.9 6.8 (0.4–81.0) | 10.4 ± 13.7 6.0 (0.04–106.0) | 1. 0.0009 2. <0.0001 3. 0.8766 |
| PSAD | 0.15 ± 0.22 0.07 (0.001–2.21) | 0.23 ± 0.28 0.12 (0.001–2.06) | 0.14 ± 0.20 0.07 (0.001–1.69) | 0.24 ± 0.29 0.13 (0.001–2.21) | 0.23 ± 0.25 0.14 (0.01–1.69) | 0.22 ± 0.28 0.1 (0.001–2.21) | 1. <0.0001 2. <0.0001 3. 0.6631 |
| mMR PIRADS score | (data on 149) | (data on 88) | (data on 113) | (data on 124) | (data on 71) | (data on 129) | 1. 0.447 2. 0.891 3. 0.393 |
| PIRADS 2 | 9 (6.05%) | 4 (4.54%) | 5 (4.43%) | 8 (6.45%) | 2 (2.82%) | 7 (5.43%) | |
| PIRADS 3 | 23 (15.44%) | 11 (12.5%) | 20 (17.70%) | 14 (11.29%) | 10 (14.08%) | 19 (14.73%) | |
| PIRADS 4 | 84 (56.36%) | 56 (63.64%) | 61 (53.98%) | 79 (63.70%) | 45 (63.38%) | 72 (55.81%) | |
| PIRADS 5 | 33 (22.15%) | 17 (19.32%) | 27 (23.89%) | 23 (18.55%) | 14 (19.72%) | 31 (24.03%) | |
| Diagnosis | | | | | | | 1. 0.021 2. 0.034 3. 0.045 |
| BPH | 327 (55.8%) | 132 (40.2%) | 289 (59.9%) | 167 (38.9%) | 52 (25.8%) | 187 (40.8%) | |
| All PC | 259 (44.2%) | 196 (59.7%) | 193 (40%) | 262 (61.%) | 149 (74.1%) | 271 (59.1%) | |
| Clinical significant PC | 205 (34.9%) | 166 (50.6%) | 149 (30.9%) | 148 (34.5%) | 127 (63.1%) | 217 (47.3%) | |
| nmPC | 220 (37%) | 135 (41.1%) | 173 (35.8%) | 182 (41.7%) | 129 (61.1%) | 198 (42.5%) | |
| mPC | 39 (7.1%) | 61 (18.5%) | 20 (4.1%) | 80 (19.3%) | 20 (9.9%) | 73 (16.5%) | |

**Table 3.** Characteristics of the non-metastatic prostate cancer (nmPC) population stratified on the basis of NLR, PLR, and AGR score. Mean ± SD, median, (range). Number of cases (%).

| Parameter | Low NLR (<2.5) | High NLR (≥2.5) | Low PLR (<120.0) | High PLR (≥120.0) | Low AGR (<1.4) | High AGR (≥1.4) | *p* Value 1. NLR 2. PLR 3. AGR |
|---|---|---|---|---|---|---|---|
| Number of cases with available ratios | 220 (62.0%) | 135 (38.0%) | 173 (67.8%) | 182 (32.2%) | 129 (39.4%) | 198 (61.6%) | / |
| Age (years) | 64.7 ± 8.7; 66.0: (40–87) | 64.9 ± 6.6 66.0: (44–78) | 65.9 ± 6.7 66.0: (47–87) | 64.0 ± 6.9 65.0: (40–78) | 66.2 ± 6.2 67.0: (48–84) | 64.4 ± 6.7 65.0: (47–81) | 1. 0.836 2. 0.010 3. 0.014 |
| Total PSA (ng/mL) | 10.1 ± 9.1; 4.3: (1.7–86.0) | 8.1 ± 4.7 7.0:(0.06–30.0) | 10.4 ± 10.3 7.4: (0.05–86.0) | 8.5 ± 6.3 7.0: (0.06–58.0) | 9.3 ± 6.7 7.7: (1.7–48.0) | 10.0 ± 10.5 7.1: (0.04–86.0) | 1. 0.022 2. 0.031 3. 0.471 |
| PSAD | 0.23 ± 0.18; 0.07: (0.02–1.56) | 0.22 ± 0.19 0.16: (0.01–1.48) | 0.23 ± 0.24 0.08: (0.01–1.56) | 0.35 ± 0.24 0.08: (0.01–2.20) | 0.24 ± 0.20 0.08: (0.04–1.48) | 0.23 ± 0.24 0.08: (0.01–1.56) | 1. 0.619 2.<0.0001 3. 0.694 |
| mMR PIRADS score | Data on 141 cases | Data on 85 cases | Data on 107 cases | Data on 111 cases | Data on 69 cases | Data on 129 cases | |
| PIRADS 2 | 7 (5%) | 3 (3%) | 3 (3%) | 6 (5%) | 1 (1%) | 7 (5%) | 1. 0.787 |
| PIRADS 3 | 20 (14%) | 11 (13%) | 18 (17%) | 11 (10%) | 10(15%) | 19 (15%) | 2. 0.256 |
| PIRADS 4 | 82 (58%) | 55 (65%) | 60 (56%) | 72 (65%) | 45(65%) | 72 (56%) | 3. 0.394 |
| PIRADS 5 | 32 (23%) | 16 (19%) | 26 (24%) | 22 (20%) | 13(19%) | 31 (24%) | |
| Prostate tumor size (mm) at mMR | 12.6 ± 5.0; 12.0: (5.0–26.0) | 12.8 ± 5.6 12: (4.0–35.0) | 13.52 ± 6.12 12: (4.0–38.0) | 13.22 ± 5.95 12: (4.0–47.0) | 12.06 ± 5.0 12: (4.0–38.0) | 13.64 ± 6.10 12.0: (4.0–47.0) | 1. 0.727 2. 0.639 3. 0.214 |
| Clinical T staging | | | | | | | 1. 0.871 |
| T2 | 196 (89%) | 118 (87%) | 153 (88%) | 161 (88%) | 116 (90%) | 171 (86%) | 2. 0.477 |
| T3a | 19 (9%) | 13 (10%) | 14 (8%) | 18 (10%) | 12 (9%) | 21 (11%) | 3. 0.349 |
| T3b | 5 (2%) | 4 (3%) | 6 (4%) | 3 (2%) | 1 (1%) | 6 (3%) | |
| Clinical N staging | | | | | | | 1. 0.357 |
| N0 | 205 (93%) | 129 (96%) | 163 (94%) | 171 (94%) | 120 (93%) | 187 (94%) | 2. 0.916 |
| N1 | 15 (7%) | 6 (4%) | 10 (6%) | 11 (6%) | 9 (7%) | 11 (6%) | 3. 0.600 |
| Biopsy outcomes | | | | | | | |
| % positive samples PC | 39.7 ± 25.9; 28.0: (12.0–100.0) | 30.4 ± 23.9; 28.0: (4.0–87.0) | 30.1 ± 19.2; 25: (12.0–100.0) | 41.4 ± 30.7; 32.5: (4.0–100.0) | 30.2 ± 23.2; 25.0: (4.0–95.0) | 35.3 ± 24.8; 32.0 (4.0–100.0) | 1. 0.0008 2.<0.0001 3. 0.062 |
| Max% PC tissue per core | 36.9 ± 24.3; 32.0: (2.0–94.0) | 34.8 ± 24.0; 30.0: (4.0–90.0) | 36.3 ± 20.2; 33.0; (4.0–83.0) | 35.8 ± 25.4; 32.0: (2.0–94.0) | 31.5 ± 20.2; 25.2: (4.0–77.0) | 38.8 ± 23.5; 35.0: (4.0–94.0) | 1. 0.427 2. 0.809 3. 0.004 |
| ISUP grading at biopsy | | | | | | | |
| 1 | 54 (24%) | 29 (22%) | 44 (25%) | 39 (21%) | 21 (16%) | 54 (27%) | |
| 2 | 95 (43%) | 53 (39%) | 75 (43%) | 73 (40%) | 59 (46%) | 72 (36%) | 1. 0.621 |
| 3 | 37 (17%) | 31 (23%) | 31 (18%) | 37 (20%) | 26 (20%) | 38 (19%) | 2. 0.549 |
| 4 | 24 (11%) | 14 (10%) | 17 (10%) | 21 (12%) | 14 (11%) | 25 (13%) | 3. 0.140 |
| 5 | 10 (5%) | 8 (6%) | 6 (4%) | 12 (7%) | 9 (7%) | 9 (5%) | |
| Risk Class (D'Amico) | | | | | | | |
| Low risk | 54 (24%) | 32 (24%) | 42 (24%) | 44 (24%) | 24 (19%) | 50 (25%) | 1. 0.980 |
| Intermediate risk | 120 (55%) | 74 (55%) | 96 (56%) | 98 (54%) | 75 (58%) | 103 (52%) | 2. 0.917 |
| High risk | 46 (21%) | 29 (21%) | 35 (20%) | 40 (22%) | 30 (23%) | 45 (23%) | 3. 0.354 |
| Radical prostatectomy | Data on 203 cases | Data on 124 cases | Data on 154 cases | Data on 173 cases | Data on 120 cases | Data on 187 cases | 1. 0.696 |
| Laparoscopic | 120 (59%) | 76 (61%) | 93 (60%) | 103 (60%) | 91 (76%) | 101 (54%) | 2. 0.875 |
| Robotic-assisted | 83 (41%) | 48 (39%) | 61 (40%) | 70 (40%) | 29 (24%) | 86 (46%) | 3. 0.0001 |
| Pathological stage (T) | | | | | | | |
| pT2 | 101 (50%) | 59 (48%) | 75 (49%) | 85 (49%) | 54 (45%) | 95 (51%) | 1. 0.690 |
| pT3a | 82 (40%) | 49 (39%) | 59 (38%) | 72 (42%) | 54 (45%) | 70 (37%) | 2. 0.532 |
| pT3b | 20 (10%) | 16 (13%) | 20 (13%) | 16 (9%) | 12 (10%) | 22 (12%) | 3. 0.417 |

**Table 3.** *Cont.*

| Parameter | Low NLR (<2.5) | High NLR (≥2.5) | Low PLR (<120.0) | High PLR (≥120.0) | Low AGR (<1.4) | High AGR (≥1.4) | *p* Value 1. NLR 2. PLR 3. AGR |
|---|---|---|---|---|---|---|---|
| Pathological stage (N) | | | | | | | 1. 0.457 2. 0.997 3. 0.985 |
| N0 | 191 (94%) | 119 (96%) | 146 (95%) | 164 (95%) | 113 (94%) | 176 (94%) | |
| N+ | 12 (6%) | 5 (4%) | 8 (5%) | 9 (5%) | 7 (6%) | 11 (6%) | |
| ISUP grading at surgery | | | | | | | 1. 0.103 2. 0.916 3. 0.186 |
| 1 | 37 (18%) | 19 (15%) | 29 (19%) | 27 (16%) | 16 (13%) | 37 (20%) | |
| 2 | 95 (47%) | 50 (40%) | 69 (45%) | 76 (44%) | 56 (47%) | 77 (41%) | |
| 3 | 39 (19%) | 37 (30%) | 33 (21%) | 43 (25%) | 26 (22%) | 43 (23%) | |
| 4 | 10 (5%) | 10 (8%) | 9 (6%) | 11 (6%) | 6 (5%) | 16 (9%) | |
| 5 | 22 (11%) | 8 (7%) | 14 (9%) | 16 (9%) | 16 (13%) | 14 (7%) | |
| Surgical margin (R) | | | | | | | 1. 0.841 2. 0.131 3. 0.890 |
| Negative | 151 (74%) | 91 (73%) | 108 (70%) | 134 (77%) | 89 (74%) | 140 (75%) | |
| Positive | 52 (26%) | 33 (27%) | 46 (30%) | 39 (23%) | 31 (26%) | 47 (25%) | |
| PNI at surgery | | | | | | | 1. 0.012 2. 0.365 3. 0.049 |
| Positive | 136 (67%) | 99 (80%) | 107 (69%) | 128 (74%) | 91 (76%) | 122 (65%) | |
| Negative | 67 (33%) | 25 (20%) | 47 (31%) | 45 (26%) | 29 (24%) | 65 (35%) | |
| Cribriform/IDC at surgery | | | | | | | 1. 0.740 2. 0.836 3. 0.445 |
| Positive | 39 (19%) | 22 (18%) | 28 (18%) | 33 (19%) | 24 (20%) | 31 (17%) | |
| Negative | 164 (81%) | 102 (82%) | 126 (82%) | 140 (81%) | 96 (80%) | 156 (83%) | |
| Postoperative total PSA (ng/mL) | 0.21 ± 0.96; 0.02: (0.01–10) | 0.25 ± 0.87 0.03: (0.01–7) | 0.28 ± 1.22 0.02: (0.01–10) | 0.17 ± 0.47 0.03: (0.01–2.9) | 0.13 ± 0.34 0.03: (0.01–2.34) | 0.29 ± 1.20 0.02: (0.01–10.0) | 1. 0.705 2. 0.273 3. 0.155 |
| Biochemical progression (number of cases and %) | 22 (11%) | 18 (15%) | 20 (13%) | 20 (12%) | 18 (15%) | 19 (10%) | 1. 0.247 2. 0.645 3. 0.172 |

### 3.2.1. Benign Prostatic Hyperplasia versus Prostate Cancer Diagnosis (Group A versus B)

Neutrophil-to-Lymphocyte Ratio

A total of 586 cases showed low NLR and 328 cases high NLR according to the cut-off 2.5. No significant differences ($p > 0.05$) were found in terms of age, prostate volume, and PIRADS score between the two groups, whereas in the high NLR group the mean values of total PSA were significantly ($p = 0.0009$) higher ($10.36 \pm 12.82$) when compared to low NLR ($7.63 \pm 11.33$). Cases with a low NLR showed a higher percentage of BPH diagnosis (55.8%) than cases with a high NLR (40.2%). Cases with a high NLR showed a higher percentage of clinically significant (50.6%) and metastatic PC (18.5%) than cases with low NLR (34.9% and 7.1%, respectively) (Table 2).

Platelet-to-Lymphocyte RATIO

A total of 482 cases showed low PLR and 429 cases with high PLR according to the cut-off 120.0. No significant differences ($p > 0.05$) were found in terms of age, prostate volume, or PIRADS score between the two groups, whereas in high PLR group mean values of total PSA were significantly ($p < 0.0001$) higher ($10.39 \pm 12.89$) when compared to low PLR ($6.99 \pm 10.72$). Cases with a low PLR showed a higher percentage of BPH diagnosis (59.9%) than cases with high PLR (38.9%). Cases with a high PLR showed a higher percentage of clinically significant (34.5%) and metastatic PC (19.3%) than cases with low PLR (30.9% and 4.1%, respectively) (Table 2).

Albumin-to-Globulin Ratio

A total of 201 cases showed low AGR and 458 cases high AGR according to the cut-off 1.4. No significant differences ($p > 0.05$) were found in terms of age, prostate volume, total

PSA, and PIRADS score between the two groups. Cases with a high AGR showed a higher percentage of BPH diagnosis (40.8%) than cases with a low AGR (25.8%). Cases with a low AGR showed a higher percentage of clinically significant (63.1%) but lower of metastatic PC (9.9%) than cases with high AGR (47.3% and 16.5%, respectively) (Table 2).

3.2.2. Non-Metastatic Prostate Cancer (Group B2)
Neutrophil-to-Lymphocyte Ratio

In the non-metastatic PC group, 62% showed low NLR and 38% high NLR according to the cut-off 2.5. No significant differences ($p > 0.05$) were found in terms of T staging, ISUP grading, and biochemical progression after RP. The percentage of PNI at final pathology was significantly ($p = 0.012$) higher in the high (80%) than in the low (67%) NLR group (Table 3).

Platelet-to-Lymphocyte Ratio

A total of 67.8% showed low PLR and 32.2% high PLR according to the cut-off of 120.0. No significant differences ($p > 0.05$) were found in terms of T staging, ISUP grading, other pathologic variables, and biochemical progression (Table 3).

Albumin-to-Globulin Ratio

A total of 61.6% showed high AGR and 39.4% low AGR according to the cut-off 1.4. No significant differences ($p > 0.05$) were found in terms of T stage, ISUP grading, and biochemical progression. The percentage of PNI at final pathology was significantly ($p = 0.049$) higher in the low (76%) than in the high (65%) AGR group (Table 3).

*3.3. Variation in Neutrophil-to-Lymphocyte, Platelet-to-Lymphocyte, and Albumin-to-Globulin Ratio According to Radical Prostatectomy Procedure*

A total of 371 cases with PC diagnosis were submitted to radical prostatectomy. After surgery, mean NLR, PLR, and AGR values significantly ($p < 0.0001$) varied when compared to pre-surgical values (Figure 2). In particular, NLR mean values (pre: $2.47 \pm 1.15$ and post: $10.77 \pm 6.93$) and AGR mean values (pre: $1.50 \pm 0.28$ and post: $1.65 \pm 0.20$) significantly increased, whereas PLR (pre: $128.04 \pm 49.29$ and post: $107.75 \pm 62.07$) significantly reduced after RP. These significant variations after RP were mainly maintained also after the stratification of cases on the basis of pT stage, ISUP grading, and risk classes (Table 4).

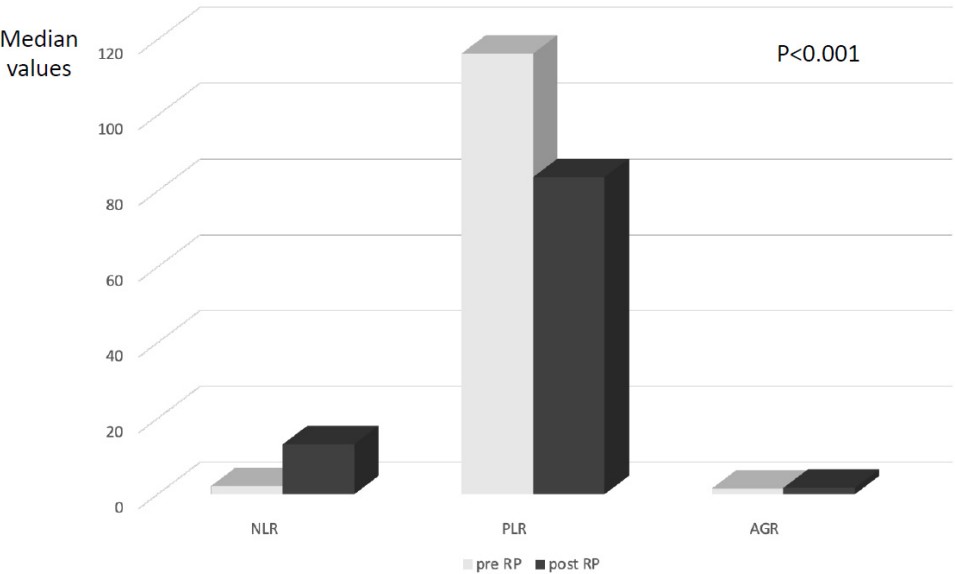

**Figure 2.** Variation in median values of NLR, PLR, and AGR after radical prostatectomy (RP) in patients with non-metastatic prostate cancer.

**Table 4.** Changes in NLR, PLR, and AGR from pre-RP to post RP determination. Mean $\pm$ SD, median, (range).

| PC Cases Submitted to RP | NLR Pre-RP | NLR Ratio Post-RP | PLR Pre-RP | PLR Post-RP | AGR Pre-RP | AGR Post-RP | *p* Value 1. NLR 2. PLR 3. AGR |
|---|---|---|---|---|---|---|---|
| All cases (371) | 2.47 ± 1.15; 2.18: (0.09–9.60) | 10.77 ± 6.93; 13.19: (1.08–26.90) | 128.04 ± 49.29; 116.31: (1.92–439.09) | 107.75 ± 62.07; 83.7: (33.13–330.0) | 1.50 ± 0.28; 1.50: (0.92–5.60) | 1.65 ± 0.20; 1.70: (1.13–2.10) | 1. <0.0001 2. <0.0001 3. <0.0001 |
| pT2 (190) | 2.52 ± 1.15; 2.18: (1.07–9.60) | 11.82 ± 6.98; 13.47: (1.08–26.95) | 128.81 ± 49.32; 116.31: (53.68–300.0) | 102.61 ± 61.83; 83.70: (46.90–261.90) | 1.49 ± 0.28; 1.50: (0.92–2.11) | 1.59 ± 0.20; 1.70: (1.16–2.0) | 1. <0.0001 2. <0.0001 3. 0.0001 |
| pT3 (181) | 2.41 ± 1.15; 2.18: (30.09–6.11) | 9.0 ± 6.93; 13.19: (1.50–22.0) | 126.77 ± 49.26; 116.31: (1.92–439.09) | 113.20 ± 62.07; 83.76: (33.13–330.0) | 1.50 ± 0.28; 1.50: (0.98–5.60) | 1.63 ± 0.20; 1.7: (1.13–2.10) | 1. <0.0001 2. 0.0218 3. <0.0001 |
| ISUP 1–2 (227) | 2.48 ± 1.15; 2.18: (0.90–9.60) | 12.09 ± 6.98; 13.47: (1.08–26.90) | 126.79 ± 49.32; 116.31: (40.95–439.09) | 98.68 ± 61.83; 83.78: (46.90–261.90) | 1.49 ± 0.28; 1.50: (0.92–5.60) | 1.62 ± 0.18; 1.7: (1.50–2.0) | 1. <0.0001 2. <0.0001 3. <0.0001 |
| ISUP 3–5 (144) | 2.43 ± 1.15; 2.18: (0.09–5.78) | 8.69 ± 6.97; 13.16: (1.50–22.03) | 129.31 ± 49.26; 116.31: (33.13–330.0) | 118.82 ± 62.07; 83.74: (33.13–330.0) | 1.50 ± 0.28; 1.50: (0.98–2.19) | 1.60 ± 0.20; 1.7: (1.13–2.10) | 1. <0.0001 2. 0.1133 3. 0.0006 |
| Low risk (90) | 2.73 ± 1.15; 2.18: (1.24–9.60) | 11.25 ± 6.95; 13.47: (1.47–26.95) | 128.21 ± 49.36; 116.31: (71.15–300.0) | 92.12 ± 62.90; 82.85: (33.13–228.86) | 1.46 ± 0.28; 1.50: (0.92–2.04) | 1.62 ± 0.20; 1.70: (1.60–2.10) | 1. <0.0001 2. <0.0001 3. <0.0001 |
| Intermediate risk (197) | 2.39 ± 1.17; 2.12: (0.77–7.45) | 10.03 ± 7.02; 13.19: (1.08–23.24) | 126.37 ± 49.68; 115: (40.95–439.09) | 117.10 ± 61.83; 83.78: (46.94–330.0) | 1.50 ± 0.28; 1.50: (0.98–5.60) | 1.54 ± 0.20; 1.70: (1.50–1.80) | 1. <0.0001 2. 0.1017 3. 0.1036 |
| High risk (84) | 2.32 ± 1.22; 2.09: (0.09–5.78) | 10.91 ± 7.04; 13.77: (1. 78–22.03) | 129.38 ± 50.16; 112.20: (1.92–371.0) | 100.20 ± 55.73; 82.85: (58.86–209.0) | 1.50 ± 0.29; 1.50: (0.98–2.19) | 1.56 ± 0.20; 1.70: (1.13–2.0) | 1. <0.0001 2. 0.0005 3. 0.1204 |

### 3.4. Correlation among Neutrophil-to-Lymphocyte, Platelet-to-Lymphocyte, Albumin-to-Globulin Ratio, and Other Clinical and Pathological Variables

We investigated significant correlations among each ratio and the clinical and pathological variables of our population, as described in Table 5. A significant correlation was found between NLR and PLR values (r = 0.590765385; $p < 0.0001$), but not between NLR or PLR and AGR ($p > 0.05$).

**Table 5.** Pearson correlation coefficients among NLR, PLR, AGR, and the different pathological and clinical variables.

| Correlation | Coefficient | *p* Value |
|---|---|---|
| NLR–PLR | 0.590765385 | **<0.0001** |
| NLR–AGR | −0.056808466 | 0.233 |
| PLR–AGR | −0.032381744 | 0.495 |
| NLR–age | −0.003393898 | 0.949 |
| NLR BMI | −0.043542716 | 0.360 |
| NLR metabolic syndrome | −0.011263298 | 0.814 |
| NLR–prostate volume | 0.012090362 | 0.798 |
| NLR–risk class | −0.031727259 | 0.509 |
| NLR–preoperative PSA | 0.060700932 | 0.196 |
| NLR–PSAD | 0.084847705 | 0.070 |
| NLR–PIRADS score | −0.019247647 | 0.686 |
| NLR–diagnosis PC | 0.119796458 | **0.000324** |
| NLR–prostate tumor size | −0.057921656 | 0.224 |
| NLR–percentage positive core at biopsy | −0.117607299 | **0.012** |

**Table 5.** *Cont.*

| Correlation | Coefficient | *p* Value |
| --- | --- | --- |
| NLR–T stage | −0.0530888 | 0.259 |
| NLR–N stage | −0.045416757 | 0.338 |
| NLR–M stage | 0.089695182 | 0.056 |
| NLR–ISUP grading | 0.022020966 | 0.639 |
| NLR–surgical margins | −0.013402868 | 0.782 |
| NLR–PNI | 0.065605296 | 0.162 |
| NLR–cribriform/IDC | −0.041355254 | 0.382 |
| NLR–postoperative PSA | 0.00894501 | 0.849 |
| NLR–biochemical progression | −0.009564575 | 0.848 |
| PLR–age | −0.041847787 | 0.388 |
| PLR–BMI | −0.095185293 | **0.045** |
| PLR–metabolic syndrome | −0.045276738 | 0.344 |
| PLR–prostate volume | −0.0306643 | 0.528 |
| PLR–risk class | 0.12076586 | **0.010** |
| PLR–preoperative PSA | 0.112118686 | **0.018** |
| PLR–PSAD | 0.138732191 | **0.003** |
| PLR–PIRADS score | −0.006489868 | 0.899 |
| PLR–diagnosis PC | 0.16085218 | **<0.0001** |
| PLR–prostate tumor size | −0.055344287 | 0.247 |
| PLR–percentage positive core at biopsy | 0.153215834 | **0.0012** |
| PLR–T stage | −0.0143615 | 0.768 |
| PLR–N stage | −0.022325927 | 0.643 |
| PLR–M stage | 0.186703018 | **0.000076** |
| PLR–ISUP grading | 0.00920659 | 0.846 |
| PLR–surgical margins | −0.031205866 | 0.514 |
| PLR–PNI | 0.078362099 | 0.099 |
| PLR–cribriform/IDC | −0.018649791 | 0.705 |
| PLR–postoperative PSA | −0.004990327 | 0.933 |
| PLR–biochemical progression | −0.001641053 | 0.983 |
| AGR–age | −0.189983803 | **0.000096** |
| AGR–BMI | −0.039096382 | 0.424 |
| AGR–metabolic syndrome | −0.054469657 | 0.268 |
| AGR–prostate volume | −0.031335926 | 0.525 |
| AGR–risk class | 0.025717996 | 0.599 |
| AGR–preoperative PSA | −0.039384808 | 0.424 |
| AGR–PSAD | −0.046284517 | 0.346 |
| AGR–PIRADS score | 0.001454159 | 0.977 |
| AGR–diagnosis PC | −0.051833904 | 0.195 |
| AGR–prostate tumor size | 0.070965303 | 0.146 |
| AGR–percentage positive core at biopsy | −0.117692213 | **0.016** |

**Table 5.** *Cont.*

| Correlation | Coefficient | *p* Value |
|---|---|---|
| AGR–T stage | 0.041302814 | 0.397 |
| AGR–N stage | 0.026190226 | 0.593 |
| AGR–M stage | −0.023100814 | 0.637 |
| AGR–ISUP grading | 0.013429298 | 0.783 |
| AGR–Risk classes | 0.025717996 | 0.599 |
| AGR–surgical margins | −0.027616388 | 0.580 |
| AGR–PNI | −0.056143388 | 0.251 |
| AGR–cribriform/IDC | −0.050339474 | 0.306 |
| AGR–postoperative PSA | 0.05732158 | 0.240 |
| AGR–biochemical progression | −0.020713105 | 0.682 |

The bold in this table can be useful to underline the significant values of *p*-value.

NLR significantly correlated with PC initial diagnosis (r = 0.119796458; *p* = 0.000324), but not with T, N, and M staging, ISUP grading or biochemical progression (*p* > 0.05) (Table 5).

PLR significantly correlated with PC initial diagnosis (r = 0.16085218, *p* < 0.0001) and M stage (r = 0.186703018; *p* = 000076), but not with T or N staging, ISUP grading, or biochemical progression (*p* > 0.05) (Table 5).

No significant (*p* > 0.05) correlations were found between *AGR* and the other variables (Table 5).

*3.5. Sensitivity, Specificity, Positive Predictive Value, Negative Predictive Value, and Area under the Curve Results in Predicting Pathologic Features*

The analysis was performed considering the optimal cut-off for each ratio of 2.5, 120.0, and 1.4, respectively, for NLR, PLR, and AGR, so as to distinguish low and high rate cases in each group.

3.5.1. Initial Diagnosis of Clinically Significant Prostate Cancer

The performance of the three ratios in predicting clinically significant PC (csPC) at initial diagnosis is reported in Table 6 (A). In our population, PLR cut-off showed the highest sensitivity (0.848) but the lowest specificity (0.256) when compared to NLR (0.000 and 1.000, respectively, for sensitivity and specificity) and AGR (0.087 and 0.974, respectively, for sensitivity and specificity). Accuracy in predictive value was higher using PLR (0.718) when compared to NLR (0.220) and AGR (0.247) and the ROC curves for AUC for the three ratios were similar (Figure 3). Considering a PPV and NPV of 0.801 and 0.323, respectively; 80% of cases with high PLR presented a csPC at initial diagnosis and 32% of cases with low PLR were negative for PC.

**Table 6.** A: sensitivity, specificity, positive predictive value (PPV), negative predictive value (NPV), accuracy, and AUC of the different ratios in predicting diagnosis of clinically significant PC. B: sensitivity, specificity, positive predictive value (PPV), negative predictive value (NPV), accuracy, and AUC of the different ratios in predicting the metastatic M+ stage. C: sensitivity, specificity, positive predictive value (PPV), negative predictive value (NPV), accuracy, and AUC of the different ratios in predicting extracapsular T3 stage. D: sensitivity, specificity, positive predictive value (PPV), negative predictive value (NPV), accuracy, and AUC of the different ratios in predicting ISUP 3–5 grading. E: sensitivity, specificity, positive predictive value (PPV), negative predictive value (NPV), accuracy, and AUC of the different ratios in predicting biochemical progression after RP.

| | Sensitivity (CI 95% Range) | Specificity (CI 95% Range) | PPV (CI 95% Range) | NPV (CI 95% Range) | Accuracy | AUC (CI 95% Range) |
|---|---|---|---|---|---|---|
| **A** | | | | | | |
| **NLR ≥ 2.5** | 0.000 (0.000–0.033) | 1.000 (0.891–1.000) | 0.000 | 0.220 | 0.220 | 0.402 (0.307–0.497) |
| **PLR ≥ 120.0** | 0.848 (0.777–0.899) | 0.256 (0.145–0.413) | 0.801 | 0.323 | 0.718 | 0.493 (0.388–0.599) |
| **AGR ≤ 1.4** | 0.087 (0.061–0.122) | 0.974 (0.902–0.998) | 0.938 | 0.190 | 0.247 | 0.458 (0.388–0.528) |
| **B** | | | | | | |
| **NLR ≥ 2.5** | 0.500 (0.217–0.783) | 0.922 (0.896–0.942) | 0.085 | 0.992 | 0.916 | 0.616 (0.332–0.900) |
| **PLR ≥ 120.0** | 0.625 (0.304–0.862) | 0.815 (0.781–0.846) | 0.047 | 0.993 | 0.813 | 0.669 (0.429–0.908) |
| **AGR ≤ 1.4** | 0.785 (0.690–0.857) | 0.321 (0.284–0.361) | 0.159 | 0.901 | 0.386 | 0.492 (0.436–0.548) |
| **C** | | | | | | |
| **NLR ≥ 2.5** | 0.574 (0.455–0.684) | 0.581 (0.467–0.687) | 0.557 | 0.597 | 0.577 | 0.483 (0.386–0.580) |
| **PLR ≥ 120.0** | 0.471 (0.357–0.588) | 0.622 (0.507–0.723) | 0.533 | 0.561 | 0.549 | 0.493 (0.397–0.589) |
| **AGR ≤ 1.4** | 0.201 (0.146–0.271) | 0.880 (0.817–0.923) | 0.640 | 0.510 | 0.531 | 0.485 (0.420–0.549) |
| **D** | | | | | | |
| **NLR ≥ 2.5** | 0.574 (0.455–0.684) | 0.581 (0.467–0.687) | 0.557 | 0.597 | 0.577 | 0.483 (0.386–0.580) |
| **PLR ≥ 120.0** | 0.471 (0.357–0.588) | 0.622 (0.507–0.723) | 0.533 | 0.561 | 0.549 | 0.493 (0.397–0.589) |
| **AGR ≤ 1.4** | 0.199 (0.144–0.268) | 0.881 (0.818–0.924) | 0.640 | 0.508 | 0.529 | 0.485 (0.421–0.549) |
| **E** | | | | | | |
| **NLR ≥ 2.5** | 0.727 (0.428–0.905) | 0.539 (0.463–0.614) | 0.095 | 0.967 | 0.551 | 0.620 (0.447–0.792) |
| **PLR ≥ 120.0** | 0.455 (0.214–0.719) | 0.873 (0.812–0.916) | 0.192 | 0.960 | 0.847 | 0.604 (0.395–0.814) |
| **AGR ≤ 1.4** | 0.231 (0.125–0.386) | 0.869 (0.832–0.900) | 0.153 | 0.917 | 0.810 | 0.479 (0.373–0.584) |

### 3.5.2. Metastatic Disease (M+)

The performance of the 3 ratios in predicting a metastatic stage is reported in Table 6 (B). In our population, NLR cut-off showed the lowest sensitivity (0.500) but the highest specificity (0.922) when compared to PLR (0.625 and 0.815, respectively, for sensitivity and specificity) and AGR (0.785 and 0.321, respectively, for sensitivity and specificity). Accuracy in predictive value was higher using NLR (0.916) when compared to PLR (0.813) and AGR (0.386) and the AUC for the three ratios were similar between NLR and PLR and higher when compared to AGR (Figure 4). Considering a PPV and NPV of 0.085 and 0.992, respectively, 8% of cases with high NLR presented a metastatic PC, and 99% of cases with low NLR presented a non-metastatic disease.

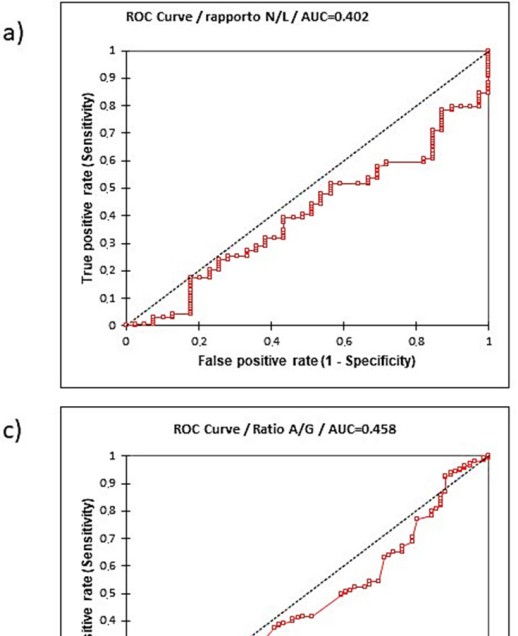
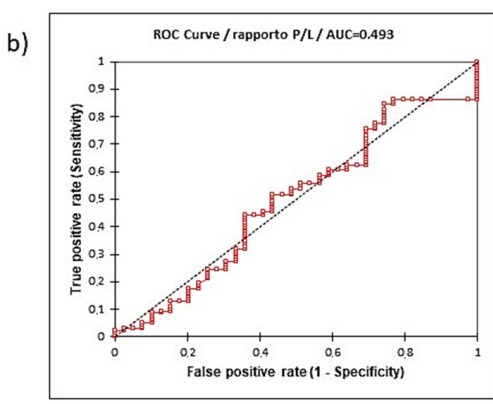
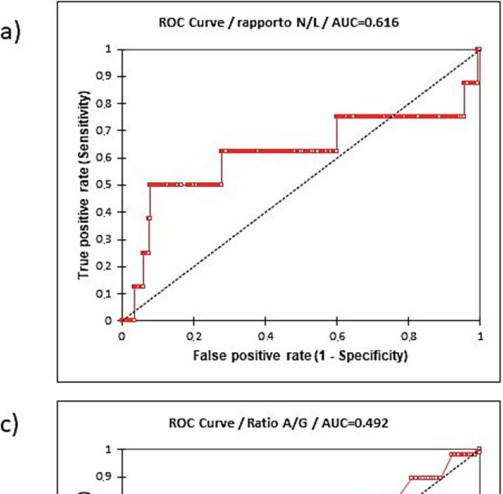

**Figure 3.** Receiver operating characteristic (ROC) curve and relative area under the curve. (AUC) of PC of NLR (**a**), PLR (**b**), and AGR (**c**) in predicting initial diagnosis.

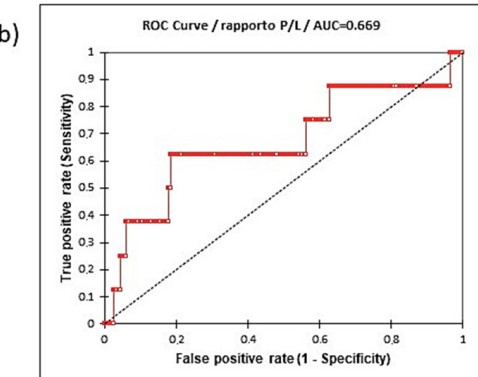
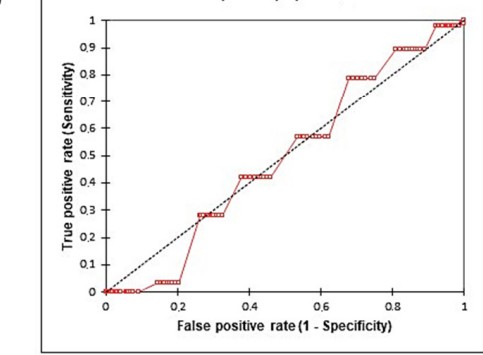

**Figure 4.** Receiver operating characteristic (ROC) curve and relative area under the curve. (AUC) of NLR (**a**), PLR (**b**), and AGR (**c**) in predicting metastatic stage.

### 3.5.3. Extraprostatic Disease (T3)

The performance of the three ratios in predicting an extracapsular T3 stage is reported in Table 6 (C). In our population, AGR cut-off showed the lowest sensitivity (0.201) but the highest specificity (0.880) when compared to PLR (0.471 and 0.622, respectively, for sensitivity and specificity) and NLR (0.574 and 0.581, respectively, for sensitivity and specificity). Accuracy in predictive value was similar among the three ratios and also the ROC curves for AUC were similar (Figure 3c). Considering a PPV and NPV of 0.640 and 0.510, respectively, 64% of cases with low AGR presented an extracapsular PC and 51% of cases with high AGR presented a T2 disease.

### 3.5.4. Aggressive Disease (ISUP 3–5)

The performance of the three ratios in predicting an aggressive ISUP 3–5 PC is reported in Table 6 (D). In our population, NLR cut-off showed the highest sensitivity (0.574) but the lowest specificity (0.581) when compared to PLR (0.471 and 0.622, respectively, for sensitivity and specificity) and AGR (0.199 and 0.881, respectively, for sensitivity and specificity). Accuracy in predictive value was higher using PLR (0.549) when compared to NLR (0.577) and AGR (0.529) and the ROC curves for AUC for the three ratios were similar (Figure 3d). Considering a PPV and NPV of 0.557 and 0.597, respectively, 38% of cases with high NLR presented a ISUP 3–5 PC, and 73% of cases with low NLR presented a ISUP 1–2 disease.

### 3.5.5. Biochemical Progression

The performance of the three ratios in predicting a biochemical progression is reported in Table 6 (E). In our population, NLR cut-off showed the highest sensitivity (0.727) but the lowest specificity (0.539) when compared to PLR (0.455 and 0.873, respectively, for sensitivity and specificity) and AGR (0.231 and 0.869, respectively, for sensitivity and specificity). Accuracy in predictive value was higher using PLR (0.847) when compared to NLR (0.551) and AGR (0.810) and the ROC curves for AUC were similar between NLR and PLR and higher when compared to AGR (Table 6 (E)). Considering a PPV and NPV of 0.095 and 0.967, respectively, 38% of cases with high NLR presented a biochemical progression, and 73% of cases with low NLR remained without biochemical progression at follow-up.

### 3.6. Logistic Regression Analysis

Table 7 shows a logistic regression analysis carried out to identify the predictive value of the three ratios in comparison with other clinical and pathological variables in terms of different outcomes. At the univariate analysis, the risk of clinically significant PC at initial diagnosis significantly increased only according to PLR ($p = 0.040$) with an OR 1.646 (95% CI 1.023–2.649) (Table 7 (A)). The risk of a metastatic disease significantly increased according to all three ratios; in particular, it increased 3.2 times for an NLR > 2.5 (95% CI 2.089–4.914), 5.2 times for a PLR > 120 (95% CI 3.182–8.812), and 0.5 times for an AGR < 1.4 (95% 0.340–0.972). In the multivariate analysis, the three ratios maintained an independent predictive value in terms of risk for a metastatic PC ($p < 0.05$), together with ISUP grading ($p < 0.001$) (Table 7 (B)).

On the contrary, neither the risk of an extracapsular PC, nor of an aggressive ISUP 3–5 disease, nor of biochemical progression significantly varied according to the three ratios (Table 7 (C–E)).

**Table 7.** (**A**): logistic regression analysis to identify predictors for clinically significant prostate cancer diagnosis. Univariate and multivariate analysis. Odds ratio (OR), 95% confidential interval (CI). (**B**): logistic regression analysis to identify predictors for metastatic stage (M+) PC. Univariate and multivariate analysis. Odds ratio (OR), 95% confidential interval (CI). (**C**): logistic regression analysis to identify predictors for extracapsular T stage (T3) PC. Univariate and multivariate analysis. Odds ratio (OR), 95% confidential interval (CI). (**D**): logistic regression analysis to identify predictors for ISUP grading 3–5 PC. Univariate and multivariate analysis. Odds ratio (OR), 95% confidential interval (CI). (**E**): logistic regression analysis to identify predictors for biochemical progression after RP for PC. Univariate and multivariate analysis. Odds ratio (OR), 95% confidential interval (CI).

| **A** | | | | | |
|---|---|---|---|---|---|
| | | **Univariable** | | | |
| | | **OR** | **95% CI_Lower** | **95% CI_Upper** | ***p*-Value** |
| **Preoperative PSA (ng/mL)** | <4 | Ref | - | - | - |
| | >4 | 0.808 | 0.367 | 1.782 | 0.598 |
| **PIRADS score** | 1–3 | Ref | - | - | - |
| | 4–5 | 1.458 | 0.722 | 2.942 | 0.293 |
| **NLR** | <2.5 | Ref | - | - | - |
| | ≥2.5 | 1.545 | 0.943 | 2.532 | 0.084 |
| **PLR** | <120 | Ref | - | - | - |
| | ≥120.0 | 1.646 | 1.023 | 2.649 | 0.040 |
| **AGR** | >1.4 | Ref | - | - | - |
| | ≤1.4 variabile categorica | 1.466 | 0.853 | 2.519 | 0.166 |

| **B** | | | | | | | | | |
|---|---|---|---|---|---|---|---|---|---|
| | | **Univariable** | | | | **Multivariable** | | | |
| | | **OR** | **95% CI_Lower** | **95% CI_Upper** | ***p*-Value** | **OR** | **95% CI_Lower** | **95% CI_Upper** | ***p*-Value** |
| **Preoperative PSA (ng/mL)** | <4 | Ref | _ | _ | _ | | | | |
| | >4 | 1.411 | 0.642 | 3.102 | 0.391 | 1.135 | 1.097 | 1.174 | <0.0001 |
| **PIRADS score** | 1–3 | Ref | _ | _ | _ | | | | |
| | 4–5 | 1.712 | 0.085 | 34.589 | 0.726 | | | | |
| **NLR** | <2.5 | Ref | _ | _ | _ | | | | |
| | ≥2.5 | 3.204 | 2.089 | 4.914 | <0.0001 | 2.241 | 0.946 | 5.311 | 0.067 |
| **PLR** | <120 | Ref | _ | _ | _ | | | | |
| | ≥120.0 | 5.295 | 3.182 | 8.812 | <0.0001 | 2.717 | 1.010 | 7.307 | 0.048 |
| **AGR** | >1.4 | Ref | _ | _ | _ | | | | |
| | ≤1.4 | 0.575 | 0.340 | 0.972 | 0.039 | 0.414 | 0.190 | 0.901 | 0.026 |
| **T stage** | **T1–2** | Ref | _ | _ | _ | | | | |
| | **T3** | 1.613 | 0.915 | 2.842 | 0.098 | | | | |
| **ISUP grading** | **1–2** | Ref | _ | _ | _ | | | | |
| | **3–5** | 68.416 | 4.155 | 1126.443 | 0.003 | 3.339 | 2.222 | 5.017 | <0.0001 |

**Table 7.** *Cont.*

| C | | Univariable | | | |
|---|---|---|---|---|---|
| | | OR | 95% CI_Lower | 95% CI_Upper | *p*-Value |
| Preoperative PSA (ng/mL) | <4 | Ref | – | – | – |
| | >4 | 1.574 | 0.741 | 3.343 | 0.238 |
| PIRADS score | 1–3 | Ref | – | – | – |
| | 4–5 variabile categorica | - | - | - | - |
| NLR | <2.5 | Ref | – | – | – |
| | ≥2.5 | 1.062 | 0.680 | 1.659 | 0.790 |
| PLR | <120 | Ref | – | – | – |
| | ≥120.0 | 0.959 | 0.622 | 1.479 | 0.851 |
| AGR | >1.4 | Ref | – | – | – |
| | ≤1.4 | 1.209 | 0.767 | 1.906 | 0.414 |
| T stage | **T1–2** | Ref | – | – | – |
| | **T3** variabile categorica | - | - | - | - |
| ISUP grading | **1–2** | Ref | – | – | – |
| | **3–5** | 9.836 | 5.122 | 18.890 | <0.0001 |

| D | | Univariable | | | |
|---|---|---|---|---|---|
| | | OR | 95% CI_Lower | 95% CI_Upper | *p*-Value |
| Preoperative PSA (ng/mL) | <4 | Ref | – | – | – |
| | >4 | 0.810 | 0.390 | 1.683 | 0.572 |
| PIRADS score | 1–3 | Ref | – | – | – |
| | 4–5 variabile categorica | 1.225 | 0.605 | 2.478 | 0.573 |
| NLR | <2.5 | Ref | – | – | – |
| | ≥2.5 | 1.482 | 0.938 | 2.340 | 0.092 |
| PLR | <120 | Ref | – | – | – |
| | ≥120.0 | 1.189 | 0.760 | 1.860 | 0.447 |
| AGR | >1.4 | Ref | – | – | – |
| | ≤1.4 | 0.977 | 0.613 | 1.557 | 0.923 |
| T stage | **T1–2** | Ref | – | – | – |
| | **T3** | 8.599 | 5.278 | 14.009 | <0.0001 |
| ISUP grading | **1–2** | Ref | – | – | – |
| | **3–5** | - | - | - | - |

**Table 7.** *Cont.*

| | E | Univariable | | | |
|---|---|---|---|---|---|
| | | OR | 95% CI_Lower | 95% CI_Upper | *p*-Value |
| **Preoperative PSA (ng/mL)** | <4 | Ref | – | – | – |
| | >4 | 1.276 | 0.371 | 4.384 | 0.699 |
| **PIRADS score** | 1–3 | Ref | – | – | – |
| | 4–5 | 0.958 | 0.306 | 3.002 | 0.942 |
| **NLR** | <2.5 | Ref | – | – | – |
| | ≥2.5 | 1.156 | 0.607 | 2.202 | 0.658 |
| **PLR** | <120 | Ref | – | – | – |
| | ≥120.0 | 0.754 | 0.396 | 1.433 | 0.389 |
| **AGR** | >1.4 | Ref | – | – | – |
| | ≤1.4 | 1.611 | 0.830 | 3.129 | 0.159 |
| **T stage** | T1–2 | Ref | – | – | – |
| | T3 | 3.709 | 1.817 | 7.571 | 0.0001 |
| **ISUP grading** | 1–2 | Ref | – | – | – |
| | 3–5 | 1.851 | 1.431 | 2.394 | < 0.0001 |

## 4. Discussion

A prognostic role for NLR, PLR, or AGR has been underlined in several solid tumors [7,8]. For example, in lung cancer, a poorer OS in patients with elevated NLR or PLR values (HR 1.18 and 1.14) has been described as they were associated with a greater risk of lymph node metastases development, poor tumor differentiation, and vascular invasion [9]. The relationships between NLR and PLR with esophageal and breast cancers were also studied and a lower OS and CSS for NLR and PLR values beyond the cut-off (OS: HR 1.55 for NLR and 1.37 for PLR in esophageal cancer and 1.46 for NLR in breast cancer) has been described [10,13]. The serum albumin/globulin ratio (AGR) has been suggested as a prognostic marker for colorectal cancer, lung cancer, breast cancer, and nasopharyngeal carcinoma [13–16].

Hypoalbuminemia was also studied in relation to fibrinogen values in other neoplastic diseases, such as in muscle-invasive bladder tumors [15]. Authors showed that a low ratio was associated with poor differentiation, non-organ-confined disease, and independently predicted time to progression [15].

We recently published two meta-analyses on the prognostic role of these ratios in PC [8,17].

The first meta-analysis on AGR found a very low level of heterogeneity (I$^2$ = 7.0%) of results among studies. In non-metastatic PC cases, pretreatment AGR was not able to show a significant predictive value either in terms of pathologic features (T and N staging, ISUP grading) or in terms of biochemical progression risk. Considering a random effect model, the pooled risk difference for non-organ confined PC, lymph-node involvement, and BCP between low and high AGR groups was close to 0.00. Only one study (18) analyzed AGR in metastatic PC. In this population, significant results were obtained either in terms of PFS or CSS prediction with a maintained independent (*p* < 0.01) value for AGR at multivariate analysis. Authors (18) reported 68.0% of patients in the low AGR and 50.9% in the high AGR group experienced tumor progression and a higher percentage of cases in the low AGR (77.0%) than in the high AGR (27.2%) group who died from PC.

The second meta-analysis found a high rate of heterogeneity either among studies on PLR ($I^2$ = 71.49%; test of group differences $p < 0.001$), or on NLR ($I^2$ = 95.87%; test of group differences $p = 0.72$) regarding the analysis on tumor stage and aggressiveness, whereas a lower rate of heterogeneity ($I^2 < 50\%$; test of group differences $p > 0.05$) was present in the analysis on progression.

A low predictive value of both NLR and PLR was found in terms of T staging or PC aggressiveness. The pooled risk difference for non-organ confined PC between high NLR and low NLR cases was 0.06 (95% CI: $-0.03$–0.15) and between high PLR and low PLR increased to 0.30 (95% CI: 0.16–0.43). A higher predictive value was found in terms of risk for progression. In particular, a higher pooled HR for overall mortality in the metastatic PC population was related to a high NLR (1.79 (95% CI:1.44–2.13)) when compared to a high PLR (1.05 (95% CI:0.87–1.24)).

Despite the numerous data present in the literature, most of the works on these three ratios in PC are retrospective, and the few prospective ones lack various analytical data or do not stratify the population. The result is that the research interest in these ratios to date has not been able to transform into clinical indications in the management of PC.

The strength of our study is represented by the prospective design in a situation of normal clinical practice. Furthermore, the population considered is numerically significant, there is a control group with BPH diagnosis, and patients with PC can be stratified into two subgroups with non-metastatic and metastatic disease. In this way, our prospective analysis allows for the first time to compare the three ratios both in terms of initial diagnosis of clinically significant PC and in terms of aggressiveness and staging of tumors. Another novelty related to our study is the longitudinal analysis in patients undergoing radical prostatectomy on the changes in the three ratios induced by surgery. The major limitation of our study lies in the analysis of survival limited to the risk of biochemical progression, given the short follow-up.

As in the present literature, different analyses were performed, distinguishing patients in low and high rate cases on the basis of cut-offs determined by ROC and Youden's index [12]. The optimal cut-offs in our population were 2.5, 120.0, and 1.4 for NLR, PLR, and AGR, respectively, similar to those used in previous studies [18–21].

We also considered the ratios as continuous variables based on the mean and median values, an analysis that is often absent in the literature.

The first data to be underlined is a significant correlation between NLR and PLR values (r = 0.590765385; $p < 0.0001$), but not between NLR or PLR and AGR ($p > 0.05$).

Regarding the initial diagnosis of PC and clinically significant PC, either mean values of NLR and PLR or the percentage of cases with high NLR and PLR significantly ($p < 0.01$) increased between the group without PC and those with PC or csPC. On the contrary, no significant differences were found in terms of AGR. Accuracy in predictive value for csPC was higher using PLR (0.718) when compared to NLR (0.220) and AGR (0.247), but, despite a high sensitivity (0.849), a very low specificity (0.256) was present. A total of 80% of cases with high PLR and only 20% of those with low PLR presented a clinically significant PC at initial diagnosis and the risk of csPC significantly increased only according to PLR ($p = 0.040$) with an OR = 1.646 (95% CI: 1.023–2.649).

Regarding the predictive value of the ratios in terms of aggressiveness and T stage in non-metastatic PC, no significant differences were found either in terms of mean values or in terms of percentage of high or low ratios. The accuracy was particularly low and the risk for an extracapsular T3 disease or a ISUP 3–5 PC did not significantly increase according to none of the three ratios.

The best performance as predictive value, limited to NLR and PLR, was found for the risk of metastatic disease. The percentage of cases with metastatic PC significantly increased according to high NLR (from 7.1% to 18.5%) and high PLR (from 4.1% to 19.3%), and also mean value of PLR was significantly ($p < 0.01$) higher in metastatic (149.6 ± 49.5) than in non-metastatic (127.8 ± 49.3) cases. Accuracy was 0.916 and 0.813, respectively, for NLR and PLR cut-off, with higher specificity than sensitivity for both. In particular, for both NLR

and PLR, 99% of cases with a ratio under the cut-off presented a non-metastatic disease whereas only 8% (NLR) and 4% (PLR) of cases with a ratio over the cut-off presented a metastatic PC. The risk of a metastatic disease increased 3.2 times for an NLR > 2.5 (95% CI: 2.089–4.914) and 5.2 times for a PLR > 120 (95% CI: 3.182–8.812) and at the multivariate analysis, the ratios maintained an independent predictive value in terms of risk for a metastatic PC ($p < 0.05$), together with ISUP grading ($p < 0.001$).

Accuracy in predictive value for a biochemical progression was higher using PLR (0.631) when compared to NLR (0.553) and AGR (0.557) and 38% of cases with high NLR presented a biochemical progression and 73% of cases with low NLR remained without biochemical progression at follow-up. The risk of a biochemical progression did not significantly vary according to the ratios, results that could be negatively influenced by a limited follow-up that does not allow a complete survival analysis.

The interpretation of the results obtained through the longitudinal analysis in patients undergoing radical prostatectomy is uncertain. Radical removal of the prostate induces significant changes in all the ratios, regardless of the stratification based on stage, grading, and risk classes. However, whereas NLR underwent the greatest increase post-surgery, PLR was significantly reduced. The explanation of a significant but inverse behavior between PLR and NLR after surgery is contrary to the univocity of the results between the two ratios in relation to the other predictive analyses.

## 5. Conclusions

Our prospective study in the real world allows for the first time the comparison of the three ratios both in terms of initial diagnosis of clinically significant PC, and in terms of prognostic value for the aggressiveness and staging in a large population of non-metastatic and metastatic PC compared to BPH controls.

PLR appears to have the greatest accuracy in predicting an initial diagnosis of csPC but with very low specificity. PLR and NLR have a significant predictive value towards the development of metastatic disease but not in relation to variations in aggressiveness or T staging inside the non-metastatic PC.

These ratios can represent the inflammatory and immunity status of the patient related to several conditions other than PC. The simplicity of the analysis is certainly the major advantage of these ratios, being influenced by a large number of coexisting inflammatory conditions in the patient strongly limits their specific prognostic value for prostate cancer characteristics.

It is possible to hypothesize that in metastatic disease, the systemic involvement of the tumor causes variations in inflammatory and immune indices that can be translated into variations in the ratios, rather than the opposite process. Our results suggest an unlikely introduction of these analyses into clinical practice in support of validated PC risk predictors.

**Author Contributions:** Conceptualization, A.S. and S.S.; methodology, A.S., S.S. and M.F.; software, F.D.G.; validation, E.D.B., G.M. and A.G.; formal analysis, G.B.D.P., S.C. (Susanna Cattarino) and F.F.; investigation, P.V., P.C. and D.R.; resources, G.G. and M.F.; data curation, G.G., B.I.C. and M.L.E.; writing—original draft preparation, S.S., M.F., G.B., P.V., P.C., E.D.B., G.B.D.P., S.C. (Susanna Cattarino), S.C. (Simone Crivellaro), G.G., D.R., F.D.G., A.C., A.P., B.I.C., M.L.E., R.A., F.F., A.S., G.M. and A.G.; writing—review and editing, all authors; visualization, all authors; supervision, A.S.; project administration, S.S.; funding acquisition, none. All authors have read and agreed to the published version of the manuscript.

**Funding:** This research received no external funding.

**Institutional Review Board Statement:** The prospective study has been evaluated and approved by our Ethics Committee (ref. cod. 6732 Prot. 0355/2022).

**Informed Consent Statement:** Informed consent was obtained from all subjects involved in the study.

**Data Availability Statement:** The data presented in this study are available on request from the corresponding author. The data are not publicly available to respect patients privacy.

**Conflicts of Interest:** The authors declare no conflict of interest.

## Abbreviations

| | |
|---|---|
| RP | Radical prostatectomy |
| PC | Prostate cancer |
| NLR | Neutrophil-to-lymphocyte ratio |
| PLR | Platelet-to-lymphocyte ratio |
| AGR | Albumin-to-globulin ratio |
| EAU | European Association of Urology |
| CI | Confidence interval |
| OR | Odds ratio |
| HR | Hazard ratio |
| BCP | Biochemical progression |
| CSS | Cancer-specific survival |
| OS | Overall survival |

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
