# Peer review of "Comparative Prospective and Longitudinal Analysis on the Platelet-to-Lymphocyte, Neutrophil-to-Lymphocyte, and Albumin-to-Globulin Ratio in Patients with Non-Metastatic and Metastatic Prostate Cancer"

_curroncol, doi:10.3390/curroncol29120745_

Round 1

Reviewer 1 Report

In my opinion this prospective longitudinal clinical study is well structured and  considers easily found variables in the context of prostate cancer.

Just few suggestions and comments.

-The overall analysis was well done, but do you think one day these parameters will ever enter clinical practice? What may be the future practical significance of these indeces?

-In paragraph titles I think it is better not to use acronyms but the full name to make the reading clearer

-so, in patients undergoing prostatectomy what parameters were considered? The initial ones or after surgery?

-What other limitations do you think your study might have?

-Figure 1. I think the description in the caption should be improved

Author Response

In my opinion this prospective longitudinal clinical study is well structured and considers easily found variables in the context of prostate cancer.

Just few suggestions and comments.

Q1: The overall analysis was well done, but do you think one day these parameters will ever enter clinical practice? What may be the future practical significance of these indeces?

A1: We comment on a possible future use of these ratios in Conclusion section. We believe unlike a future introduction of these ratios in pre-existing predictors. A major usefulness can be suggested in the metastatic stage of disease

Q2: In paragraph titles I think it is better not to use acronyms but the full name to make the reading clearer

A2. All the paragraph titles with acronyms had been modified.

Q3. So, in patients undergoing prostatectomy what parameters were considered? The initial ones or after surgery?

A3. In patients who underwent radical prostatectomy, both pre-operative parameters and post-operative inflammation indices (after 3 months) were considered. All data are available in the tables and described in materials and methods section 2.3.3.

Q4. What other limitations do you think your study might have?

A4. The most important limitation of the study is the lack of a longer follow-up and of the analysis of the inflammation indices after, at least, 12 months as stated in discussion line 384-385.

Q5. Figure 1. I think the description in the caption should be improved

A5. The description in the caption of figure 1 is a simple explanation of the graphics. All other informations are available in the text paragraph 3.1.

Reviewer 2 Report

The authors Salciccia et.al evaluated the albumin/globulin ratio (AGR), neutrophil/lymphocyte ratio (NLR), and platelet/lymphocyte ratio (PLR) diagnostic and prognostic predictive value in a stratified population of prostate cancer (PC) cases. They concluded that PLR and NLR have a significant predictive value towards the development of metastatic disease but not in relation to variations in aggressiveness or T staging inside the nonmetastatic PC. They showed an unlikely introduction of these analyses into clinical practice in support of validated PC risk predictors. The studies presented are convincing and well-written but not novel and some aspects of the manuscript need clarification and justification.

1) Previous studies have reported the significance of albumin/globulin ratio (AGR), neutrophil/lymphocyte ratio (NLR) and platelet/lymphocyte ratio (PLR) diagnostic and prognostic predictive value in a stratified population of prostate cancer (PC) cases.

https://pubmed.ncbi.nlm.nih.gov/31115239/

https://pubmed.ncbi.nlm.nih.gov/34242232/

https://pubmed.ncbi.nlm.nih.gov/32669872/

https://pubmed.ncbi.nlm.nih.gov/26434851/

https://www.ncbi.nlm.nih.gov/pmc/articles/PMC9599963/

2) I would suggest authors compare the differences and novelty in their evaluations to previous studies. Weaknesses include the superficial nature of the description of the study in the introduction and the abstract does not cover all the key data presented in the manuscript. Similarly, the Discussion section also needs improvement by highlighting the essential findings and their implications.

3) I would suggest authors mention the full form of BPH in the abstract.

Author Response

The authors Salciccia et.al evaluated the albumin/globulin ratio (AGR), neutrophil/lymphocyte ratio (NLR), and platelet/lymphocyte ratio (PLR) diagnostic and prognostic predictive value in a stratified population of prostate cancer (PC) cases. They concluded that PLR and NLR have a significant predictive value towards the development of metastatic disease but not in relation to variations in aggressiveness or T staging inside the nonmetastatic PC. They showed an unlikely introduction of these analyses into clinical practice in support of validated PC risk predictors. The studies presented are convincing and well-written but not novel and some aspects of the manuscript need clarification and justification.

 Q1: Previous studies have reported the significance of albumin/globulin ratio (AGR), neutrophil/lymphocyte ratio (NLR) and platelet/lymphocyte ratio (PLR) diagnostic and prognostic predictive value in a stratified population of prostate cancer (PC) cases.

 https://pubmed.ncbi.nlm.nih.gov/31115239/

 https://pubmed.ncbi.nlm.nih.gov/34242232/

 https://pubmed.ncbi.nlm.nih.gov/32669872/

 https://pubmed.ncbi.nlm.nih.gov/26434851/

 https://www.ncbi.nlm.nih.gov/pmc/articles/PMC9599963/

I would suggest authors compare the differences and novelty in their evaluations to previous studies. Weaknesses include the superficial nature of the description of the study in the introduction and the abstract does not cover all the key data presented in the manuscript. Similarly, the Discussion section also needs improvement by highlighting the essential findings and their implications.

A1: All the studied mentioned explained the significance of inflammatory indices in prostate cancer, but they are extremely different in aim and in methodology when compared to our study.

We have already carried out two metanalysis on the topic and a precise analysis was previously done. We believe that referring to these two meta analysis as stated in Discussion is a good way to compare all previous experiences.  The aim of the study is quite different and I think that a longer and more substantial literature review may go beyond the purpose and type of analysis that we performed.

These are our two recent metanalysis about the topic:

-           https://pubmed.ncbi.nlm.nih.gov/36232828/

-           Salciccia S, Frisenda M, Bevilacqua G, et al. Prognostic role of platelet-to-lymphocyte ratio and neutrophil-to-lymphocyte ratio in patients with non-metastatic and metastatic prostate cancer: a meta-analysis and systematic review . Asian J Urol. Ahead of Print

Q2: I would suggest authors mention the full form of BPH in the abstract.

A2: Revised (“BPH” in “benign prostatic hyperplasia (BPH)” in the abstract).

Round 2

Reviewer 2 Report

The authors Salciccia et.al compared the prospective and longitudinal analysis on the platelet-to-lymphocyte, neutrophil-to-lymphocyte, and albumin-to-globulin ratio in patients with non-metastatic and metastatic prostate cancer. the authors sufficiently addressed the comments raised. The manuscript is considered suitable for publication.